# Responsible Active Learning via Human-in-the-loop Peer Study

## Abstract

Active learning has been proposed to reduce data annotation efforts by only manually labelling representative data samples for training. Meanwhile, recent active learning applications have benefited a lot from cloud computing services with not only sufficient computational resources but also crowdsourcing frameworks that include many humans in the active learning loop. However, previous active learning methods that always require passing large-scale unlabelled data to cloud may potentially raise significant data privacy issues. To mitigate such a risk, we propose a responsible active learning method, namely Peer Study Learning (PSL), to simultaneously preserve data privacy and improve model stability. Specifically, we first introduce a human-in-the-loop teacher-student architecture to isolate unlabelled data from the task learner (teacher) on the cloud-side by maintaining an active learner (student) on the client-side. During training, the task learner instructs the light-weight active learner which then provides feedback on the active sampling criterion. To further enhance the active learner via large-scale unlabelled data, we introduce multiple peer students into the active learner which is trained by a novel learning paradigm, including the In-Class Peer Study on labelled data and the Out-of-Class Peer Study on unlabelled data. Lastly, we devise a discrepancy-based active sampling criterion, Peer Study Feedback, that exploits the variability of peer students to select the most informative data to improve model stability. Extensive experiments demonstrate the superiority of the proposed PSL over a wide range of active learning methods in both standard and sensitive protection settings.

## 1 Introduction

The recent success of deep learning approaches mainly relies on the availability of huge computational resources and large-scale labelled data. However, the high annotation cost has become one of the most significant challenges for applying deep learning approaches in many real-world applications. In response to this, active learning has been intensively explored by making the most of very limited annotation budgets. That is, instead of learning passively from a given set of data and labels, active learning first selects the most representative data samples that benefit a target task the most, and then annotates these selected data samples through human annotators, i.e., a human oracle. This active learning scheme has been widely used in many real-world applications, including image classification (Li & Guo, 2013; Ranganathan et al., 2017; Lin et al., 2017; Parvaneh et al., 2022) and semantic segmentation (Top et al., 2011; Siddiqui et al., 2020; Casanova et al., 2020; Nilsson et al., 2021).

In the past decades, artificial intelligence (AI) technology has been successfully applied in many aspects of daily life, while it also raises a new question about the best way to build trustworthy AI systems for social good. This problem is even more crucial to applications such as face recognition (Parkhi et al., 2015; Liu et al., 2019; Shi & Jain, 2021) and autonomous driving (Isele et al., 2018; Chitta et al., 2021; Casas et al., 2021). However, a responsible system has not received enough attention from the active learning community. Specifically, recent active learning applications have benefited a lot from cloud computing services with not only sufficient computational resources but also crowdsourcing frameworks that include many humans in the loop. The main issue is that existing methods require accessing massive unlabelled data for active sampling, which raises a significant data privacy risk, i.e., most local end users may feel worried and hesitate to upload

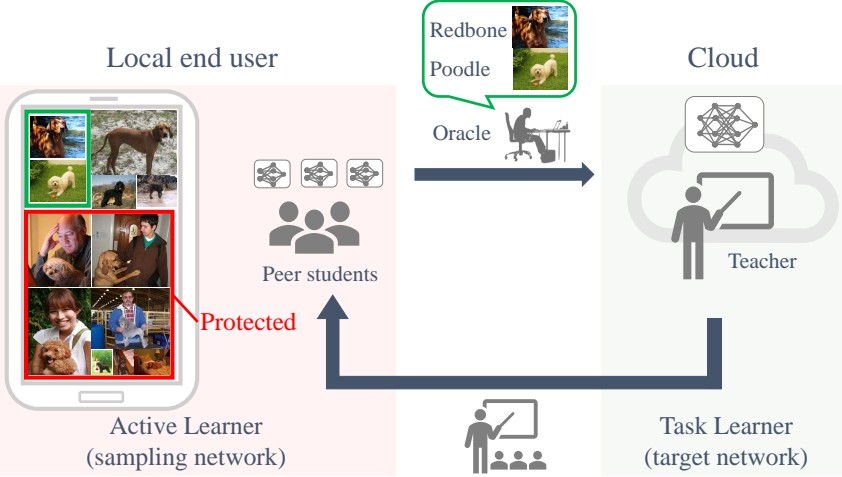

Figure 1: The overall responsible active learning framework. This framework allows the user to select a subset of protected data that can be strictly protected from anyone other than the local end user. These data may include user privacy such as face information and home decoration.

large-scale personal data from client to cloud. Therefore, it is essential to develop a responsible active learning framework that is trustworthy for local end users.

Since the main privacy risks derive from the active sampling process that requires accessing massive unlabelled data (Beluch et al., 2018; Yoo & Kweon, 2019; Ash et al., 2020; Zhang et al., 2020; Parvaneh et al., 2022), we first introduce a human-in-the-loop teacher-student architecture for data isolation that separates the model into a task learner (a target network on the cloud-side) and an active learner (a sampling network on the client-side). An intuitive example is shown in Figure 1. At present, most active sampling solutions are based on either model uncertainty (Gal et al., 2017; Beluch et al., 2018; Yoo & Kweon, 2019; Sinha et al., 2019; Zhang et al., 2020; Caramalau et al., 2021) or data diversity (Sener & Savarese, 2017; Ash et al., 2020; Parvaneh et al., 2022) i.e., data with large uncertainty or data with features span over the space will be annotated by the oracle. Nevertheless, the above-mentioned methods usually rely on representations learned by a single model, potentially making the sampling prone to bias. Sampling actively with one model means that the result highly relies on quality of the single feature extractor. And when large models are unavailable due to limited resources, the active sampling results can be unsatisfying (Coleman et al., 2019). Inspired by disagreement-based active sampling approaches (Freund et al., 1993; Cohn et al., 1994; Melville & Mooney, 2004; Cortes et al., 2019a;b), we thus introduce multiple peer students to form a robust active learner as the sampling network. For active sampling, we consider a practical setting to give the maximum protection to user data using personal album as a case study as follows: the local user has a large album with many personal photo images, where most sensitive photo images (e.g., 90% images) can be easily identified according to the tags such as timestamps and locations, and we call these images as the protected set. The objective of responsible active learning then is to select the most informative images from the remaining 10% images for annotation without uploading those 90% images to cloud. Therefore, the key to responsible active learning is to learn a proper active learner as the sampling network to train the task learner with very limited annotations.

To learn a good active learner, we introduce the proposed Peer Study Learning or PSL as follows. In real-world scenarios, all students not only learn with teacher guidance on course materials in classroom, but they also learn from each other after class using additional materials without teacher guidance. The difficulty of exercise questions in the additional materials can be estimated from the students' answers. When students cannot reach a consensus on the answer, the question is likely to be difficult and valuable. The students can provide the difficult exercise questions to the teacher as feedback. Inspired by this, the proposed Peer Study Learning consists of (1) In-Class Peer Study, (2) Out-of-Class Peer Study, and (3) Peer Study Feedback. Specifically, the In-Class Peer Study (Figure 2 (a)) allows peer students to learn from the teacher on labelled

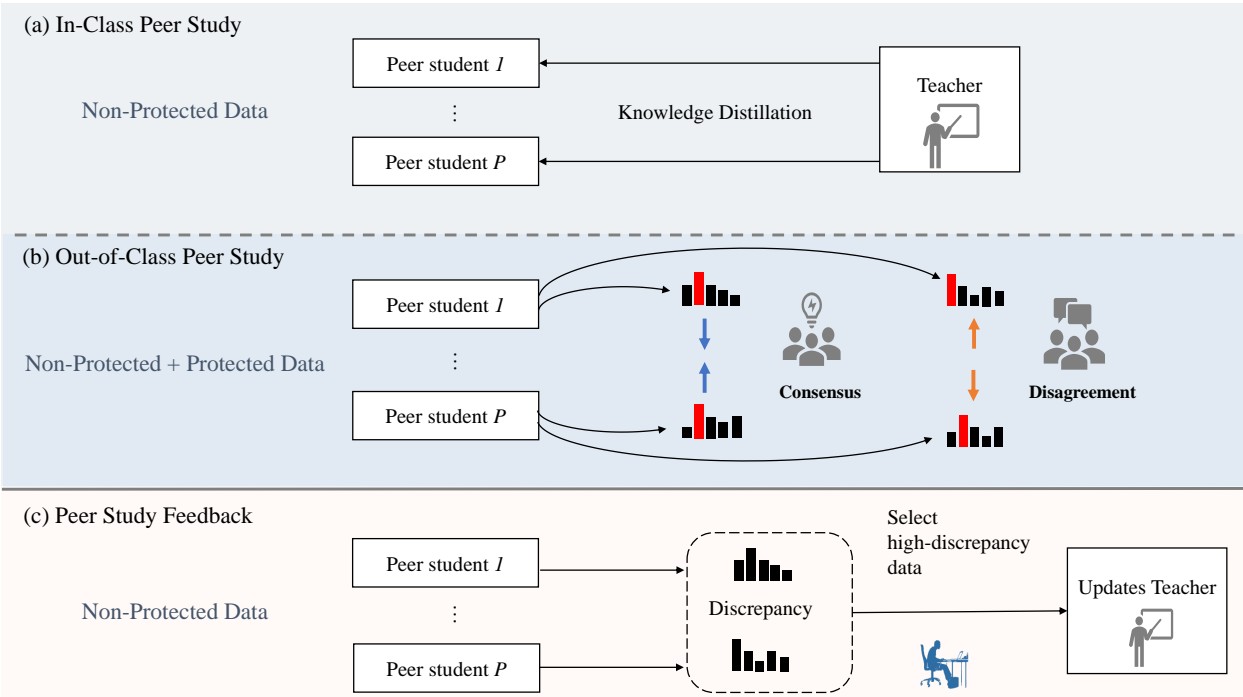

Figure 2: The main Peer Study Learning (PSL) pipeline with $P$ peer students. For simplicity, the classification loss for the teacher and peer students are omitted. For Out-of-Class Peer Study, we provide a detailed illustration of the predictions with and without consensus on the prediction scores. The last row illustrates the active sampling process based on peer study feedback.

data. In Out-of-Class Peer Study (Figure 2 (b)), the peer students collaboratively learn in a group on unlabelled data without teacher guidance. The motivation is to enable the students' cooperation on easy questions while maintaining their dissent on challenging questions. After Peer Study, we use the discrepancy of peer students as Peer Study Feedback (Figure 2 (c)). All challenging data samples that cause a large discrepancy among peer students will be selected and annotated to improve the final model performance.

In this paper, our main contributions can be summarized as follows:

- We introduce a new sensitive protection setting for responsible active learning.

- We propose a new framework PSL for responsible active learning, where a teacher-student architecture is applied to isolate unlabelled data from the task learner (teacher).

- We devise an effective learning strategy to mimic real-world in-class and out-of-class scenarios, i.e., In-Class Peer Study, Out-of-Class Peer Study, and Peer Study Feedback.

- We achieve state-of-the-art results with the proposed PSL in both standard and sensitive protection settings of active learning.

## 2   Related Work

**Active Learning.**   Recent active learning methods can be broadly divided into three distinct categories: uncertainty-based, diversity-based, and disagreement-based methods. **Uncertainty-based methods** estimate the uncertainty of predictions using metrics and select data points with uncertain predictions for query (Gal et al., 2017; Beluch et al., 2018; Yoo & Kweon, 2019; Sinha et al., 2019; Zhang et al., 2020; Wang et al., 2020; Kim et al., 2021; Caramalau et al., 2021). Instead of using the uncertainty metrics directly, some

recent works explicitly measure the informativeness of data points (Yoo & Kweon, 2019; Sinha et al., 2019; Zhang et al., 2020; Wang et al., 2020). Sinha et al. (2019) proposed to use an extra variational auto-encoder (VAE) to determine whether a data point belongs to the labelled data distribution and selects unlabelled data points that are less likely to be distributed in the labelled pool for query. This design also allows isolating the unlabelled data from the target network. Nonetheless, the large VAE-based active learner makes it costly to train and the adversarial training strategy also adds to the difficulty. **Diversity-based methods** select a set of representative and diverse data points that span the data space for query (Sener & Savarese, 2017; Ash et al., 2020; Parvaneh et al., 2022). Sener & Savarese (2017) proposed a core-set approach that treats the active sampling as a k-center problem, i.e., it selects the most representative core-set from the data pool using popular k-center algorithms. More recently, Ash et al. (2020) proposed to actively select the data points that produce gradients with diverse directions. **Disagreement-based methods** pass data points through an ensemble of hypotheses, namely a committee, and select the data points that produce the largest disagreement amongst the committee (Freund et al., 1993; Cohn et al., 1994; Melville & Mooney, 2004; Cortes et al., 2019a;b). Freund et al. (1993) proposed a simple query-by-committee method that randomly picks two hypotheses that are consistent on labelled data and predicts random unlabelled data points with them. With multiple hypotheses involved, the disagreement-based methods are less sensitive to the learning bias from individual models and are thus more stable. Previous disagreement-based active sampling methods are mainly applied on classical machine learning models, such as decision tree and random forest, on small datasets, while directly applying these methods in deep neural networks result in a huge computational cost. Our Peer Study Learning uses a set of light-weight student models to build the active learner so as to reduce the cost, where we distil knowledge from the task learner to compensate possible performance drop.

**Teacher-Student Architectures.** A teacher-student architecture has been widely applied in knowledge distillation (Hinton et al., 2015; You et al., 2017; Tian et al., 2019; Zhang et al., 2019; Mirzadeh et al., 2020), transfer learning (You et al., 2018; Furlanello et al., 2018; Yang et al., 2019; Dong & Yang, 2019; Xiang & Ding, 2020), and other tasks (Cai et al., 2019; Huo et al., 2021; Tang et al., 2021). Specifically, Hinton et al. (2015) proposed a vanilla knowledge distillation method which uses a large pretrained teacher model to guide the learning of a small student model. To learn a good compressed student, You et al. (2017; 2018) proposed to distil knowledge from multiple teachers and train multiple students. Tang et al. (2021) proposed Humble teacher where a teacher ensemble guides the student on strongly-augmented unlabelled data for semi-supervised object detection. Different from previous methods, our Peer Study Learning adopts a human-in-the-loop structure for active learning. A set of peer students benefit from not only teacher guidance via In-Class Peer Study but also the collaborative learning via Out-of-Class Peer Study using unlabelled data. The task learner (teacher) further benefits from the representative data samples selected by the active learner (student). Therefore, the teacher and peer students can be enhanced in such a responsible active learning loop.

## 3 Peer Study Learning

In this section, we first provide an overview of responsible active learning. We then introduce different components of the proposed PSL framework in detail.

### 3.1 Overview

In active learning, we have a labelled data pool $\mathcal{D}_L = \{(x_i, y_i)\}_{i=0}^{N_L}$, and an unlabelled data pool $\mathcal{D}_U = \{x_i\}_{i=0}^{N_U}$. During learning, a batch of data $\mathcal{X} = \{x_i\}_{i=0}^{b}$ in $\mathcal{D}_U$ are actively sampled to be annotated by an oracle. The annotated data is then added to the labelled pool $\mathcal{D}_L$ and removed from the unlabelled pool $\mathcal{D}_U$. The key to active learning is to identify the most informative data samples in $\mathcal{D}_U$ for annotation. In this paper, we consider a more general setting to protect the user data as follows. For active sampling, we define a safe sampling pool $\mathcal{D}_S \subseteq \mathcal{D}_U$, i.e., all images belonging to the protected set $\mathcal{D}_U \setminus \mathcal{D}_S$ cannot be uploaded to cloud in any situation. Previous works usually adopt $\mathcal{D}_S = \mathcal{D}_U$, while we allow the local end users to decide the sampling pool $\mathcal{D}_S$ in a customized way such that it can be a small subset of $D_U$. To protect user data by isolating unlabelled data from cloud, we propose Peer Study Learning with an active learner on the client-side and a task learner on the cloud-side. The active learner consists of $P$ peer students,

and is parametrized with $\theta_p = \sum_{j=0}^{P} \theta_{p_j}$. the task learner is parametrized with $\theta_T$. During training, $\theta_T$ can only be updated with labelled data, $\mathcal{D}_L$, while $\theta_p$ can be updated with both labelled and unlabelled data, $\mathcal{D}_L$ and $\mathcal{D}_U$. The main PSL pipeline is shown in Figure 2. In this task, we aim to obtain a powerful task learner with the assistance of cloud services for a local end-user that cannot access much training resources while keeping the annotation budget controllable. The large-scale task learner is placed on the cloud due to the insufficient local resources. Meanwhile, the light active learner locates locally to avoid violation of data privacy caused by uploading all unlabelled data to the cloud for active learning. As illustrated in Figure 1, the active learner samples unlabelled data locally and send them to the oracle for annotation. The annotated data are sent to the cloud to train the task learner.

### 3.2 In-Class Peer Study

In this subsection, we introduce the In-Class Peer Study process where the peer students learn from the teacher with labelled data. Since the task learner acts as both the target network and the teacher in Peer Study, we need to ensure it is well-learned. To train the task learner, we formulate the task learner's objective as:

$$\mathcal{L}_{\text{task}}(\theta_T; x_i) = \sum_{i=1}^{N} H(y_i, \sigma(f_{\theta_T}(x_i))) \tag{1}$$

where $(x_i, y_i) \in \mathcal{D}_L$ is the input data and label, $\sigma$ is the softmax function, and $f_{\theta_T}$ stands for the task model. Function $H(\cdot)$ stands for cross-entropy loss, $H(p, q) = q \log(p)$ in the image-level classification task. In the pixel-level segmentation task, it represents the pixel-wise cross-entropy loss $H(p, q) = \sum_{\Omega} q \log(p)$, where $\Omega$ stands for all pixels.

In real-world classrooms, the teacher usually provides the students with questions and answers. The students can learn from the provided material with both the teacher's explanations and the solutions in the material. Analogously, we design to let our peer students learn from both the teacher's predictions and the data labels. We accordingly formulate the In-Class Peer Study loss as:

$$\mathcal{L}_{\text{in}}(\theta_p; x_i) = \sum_{j=1}^{P} \left[ (1-\alpha)H\left(\sigma(f_{\theta_{p_j}}(x_i)), y_i\right) + \alpha\tau^2 H\left(\sigma(\frac{f_{\theta_T}(x_i)}{\tau}), \sigma(\frac{f_{\theta_{p_j}}(x_i)}{\tau})\right) \right], \tag{2}$$

where $(x_i, y_i) \in \mathcal{D}_L$ is the input data and label, the first term stands for the standard cross entropy classification loss using ground-truth labels and the second term stands for standard knowledge distillation loss using the task model outputs as the soft targets. $\alpha$ is a weight balancing the two terms. $\tau$ is the softmax temperature, larger $\tau$ implies softer predictions over classes. $f_{\theta_{p_j}}$ stands for a peer model with parameters $\theta_{p_j}$.

### 3.3 Out-of-Class Peer Study

Motivated by that the students in real-world scenarios can learn from each other after class through a group study, we conduct Out-of-Class Peer Study using unlabelled data without teacher guidance. We define the set of consistent data if the predictions from any two peers reach a consensus as follows,

$$\mathcal{S}_A = \{x : \exists p_i, p_j \ s.t. \ \arg\max(f_{\theta_{p_i}}(x)) = \arg\max(f_{\theta_{p_j}}(x)), x \in \mathcal{D}_U\}. \tag{3}$$

And for the set of inconsistent data that the predictions from all peers do not agree, we define it as follows,

$$\mathcal{S}_D = \{x : \forall p_k, p_j \ s.t. \ \arg\max(f_{\theta_{p_k}}(x)) \neq \arg\max(f_{\theta_{p_j}}(x)), x \in \mathcal{D}_U\}. \tag{4}$$

To adapt to pixel-level prediction tasks, we simply state that consensus is reached when more than half of the pixels have consistent predictions for any two peer students, and thus the datum belongs to consistent data. Otherwise, the datum belongs to inconsistent data. Then we formulate the Out-of-Class Peer Study loss as:

$$\mathcal{L}_{\text{out}}(\theta_p; x_A, x_D) = max(0, -\mathbb{1}(d_D, d_A)(d_D - d_A) + \xi)$$
$$s.t. \ \mathbb{1}(d_D, d_A) = \begin{cases} 1, & \text{if } d_D > d_A \\ -1, & \text{otherwise} \end{cases} \tag{5}$$

---

**Algorithm 1:** Peer Study for Active Learning

---

**1** **Initialize:** active learner $\theta_p$, task learner $\theta_T$
**2** **Hyperparameters:** $\alpha$, $\tau$, $\xi$
**3** **Input:** sample $(X_L, Y_L)$ from labelled pool $\mathcal{D}_L$, $(X_U)$ from unlabelled pool $\mathcal{D}_U$
**4** **for** $e = 1$ to epochs **do**
**5**    **In-class Peer Study:**
**6**    Update the task learner with equation 1
**7**    Update the active learner with equation 2
**8**    **Out-of-class Peer Study:**
**9**    Collect $\mathcal{S}_A$, $\mathcal{S}_D$ from $X_U$ with equation 3 and equation 4
**10**    Sample $(x_A, x_D)$ from $(S_A, S_D)$
**11**    Update active learner with equation 5
**12** **end for**
**13** **return** Trained $\theta_p$ and $\theta_T$
**14** **run Peer Study Feedback** to sample actively.

---

where $d = \sum_{k \neq j}^{P} KL[\sigma(f_{\theta_{p_j}}(x)||\sigma(f_{\theta_{p_k}}(x)))]$ is the model discrepancy. $d_D$ is the model discrepancy of a datum $x_D$ randomly sampled from $\mathcal{S}_D$, and $d_A$ is the model discrepancy of a datum $x_A$ randomly sampled from $\mathcal{S}_A$. $\xi$ is the margin for the Out-of-Class Peer Study loss. For pixel-level prediction, $d$ is further summed over pixels of data. The Out-of-Class Peer Study loss re-ranks unlabelled data and pushes data with different label predictions on peers to have higher priority in active sampling. The simplified form of this loss is $\mathcal{L}_{\text{out}}(\theta_p; x_A, x_D) = max(0, -|d_D - d_A| + \xi)$. Combining this with our sampling criterion Peer Study Feedback, we are able to consider both the soft prediction scores and the hard prediction labels while conducting active sampling.

### 3.4 Peer Study Feedback

In real-life, even for students learning in the same class, the learning outcome for them can still vary a lot since they have different characters or different backgrounds. Therefore, in our In-Class Peer Study, despite learning from the same teacher, the $P$ peer students' predictions may vary due to their independent initialization. And our Out-of-Class Peer Study further adjusts the prediction discrepancy among the students so that more challenging data can produce a larger discrepancy among students. Therefore, we can use the discrepancy among students to build our sampling strategy. The proposed method only uses the active learner to conduct the sample selection. Data from the unlabelled pool $\mathcal{D}_U$ are passed through our peer students and the discrepancy for any datum $x_i$ among the peers is calculated according to the following selection criterion:

$$a(\theta_p, \mathcal{D}_U) = \sum_{k \neq j}^{P} KL[\sigma(f_{\theta_{p_j}}(x_i)||\sigma(f_{\theta_{p_k}}(x_i)))]. \tag{6}$$

The calculated discrepancies of data then serve as the sampling scores. We sort the sampling scores and select the top $b$ data with largest sampling scores to send to the oracle for annotation and then add to the labelled pool $\mathcal{D}_L$. Note that in sensitive protection setting, we replace $\mathcal{D}_U$ with $\mathcal{D}_S$ in equation 6, only the non-protected data can be added to $\mathcal{D}_L$. We summarize the whole learning process for Peer Study in Alg. 1.

### 3.5 Discussion on Responsible Active Learning and Federated Learning

When it comes to privacy-preserving and responsible learning, federated learning has always been one of the solutions. In the federated learning scenario (McMahan et al., 2017), all data are stored locally and metadata like model parameters or gradients can be transferred between the server and the local device. However, transferring metadata can sometimes be risky in privacy protection. Model inversion attacks (Fredrikson et al., 2015; Wang & Kurz, 2022), for example, can extract training data using these metadata. Thus, conventional federated learning approaches do not completely isolate specific local information from the

server. Here, we consider more strict privacy preserving setting for responsible active learning. The proposed PSL keeps privacy data strictly untouched by the cloud side, not even through model parameters or metadata. Federated learning can avoid uploading raw data, while our PSL for responsible active learning can strictly protect certain extremely sensitive data with the cost of sharing a small amount of insensitive data with the task learner. Moreover, it is possible to extend our responsible approach with federated learning scenario. We show the design and results of this setting in Section 4.5.

## 4 Experiments

In this section, we first introduce a sensitive protection setting for responsible active learning. We then evaluate PSL in both sensitive protection and standard settings on image classification and semantic segmentation tasks. Lastly, we perform comprehensive ablation studies to better understand the proposed method.

### 4.1 Sensitive Protection Setting

We consider a sensitive protection setting in real-world scenarios as follows. A large portion of unlabelled data are protected by users to prevent from sending them to cloud. Data in the non-protected set can be selected and annotated by the oracle, then used to train the task learner. To simulate such a sensitive protection setting, we randomly select 90% data as the protected set, then the training starts with a labelled pool containing $N_L$ images from the non-protected set. In each step, a batch of $b$ data samples are selected by our active learner from the **unlabelled non-protected** set and sent to the oracle for annotation. The newly labelled data are then added to the labelled pool for next training steps. The above mentioned sampling process is terminated after the number of images in the labelled pool reaches a final $N_f$. A group of $P$ peer students are used as the active learner to select data for annotation.

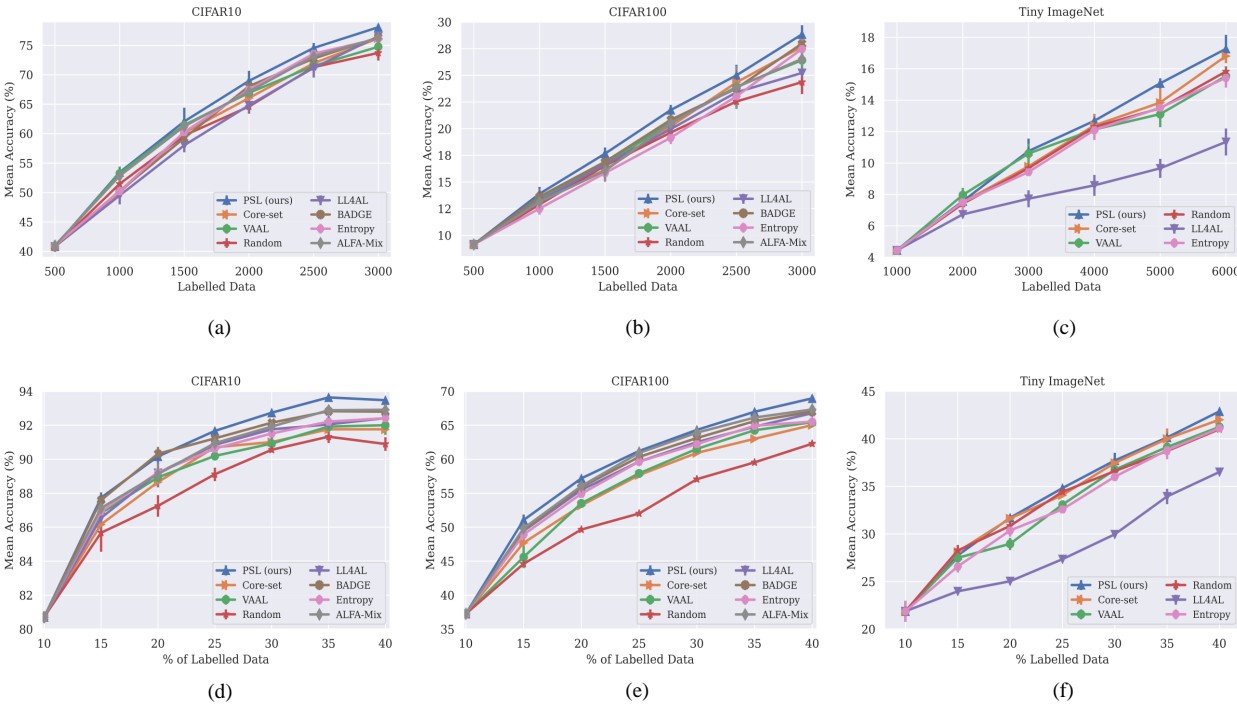

Figure 3: (a)-(c) Active learning in the sensitive protection setting on CIFAR10/100, and Tiny ImageNet. The x-axis shows the counts of labelled data. (d)-(f) Active learning in the standard setting on CIFAR10/100, and Tiny ImageNet. The x-axis shows the percentages of labelled data in all training data. The error bars represent the standard deviations among results from multiple different runs.

Table 1: A comparison of model complexity between PSL and others. 'CLS' and 'SEG' indicate the sampling models used for image classification and semantic segmentation, respectively.

| Model | CLS | SEG |
|---|---|---|
| Entropy (Shannon, 1948) | 11.17M | - |
| BADGE (Ash et al., 2020) | 11.17M | - |
| ALFA-Mix (Parvaneh et al., 2022) | 11.17M | - |
| Core-set (Sener & Savarese, 2017) | 11.17M | 15.90M |
| LL4AL (Yoo & Kweon, 2019) | 11.29M | 16.02M |
| VAAL (Sinha et al., 2019) | 23.11M | 23.11M |
| PSL (ours) | **0.16M** | **3.25M** |

## 4.2 Implementation Details

For **image classification**, we evaluate PSL on CIFAR10/100 (Krizhevsky, 2009), containing $50,000$ training images and $10,000$ testing images from 10 and 100 different classes, respectively. The labelled pool is initialized with $N_L = 500$ and we sample $b = 500$ images at each step until $N_f = 3,000$. To evaluate PSL on data with more categories, we perform experiments on Tiny ImageNet (Le & Yang, 2015) which contains $100,000$ images for training and $10,000$ images for evaluation from 200 different classes. The labelled pool is initialized with $N_L = 1,000$ and we sample $b = 1,000$ images at each step until $N_f = 6,000$. We use ResNet-18 (He et al., 2016a) for the task learner and ResNet-8 (He et al., 2016b) for the active learner. Accuracy is used as the evaluation metric. We train all models for 350 epochs with the weight $\alpha = 0.1$, softmax temperature $\tau = 4$, and the margin $\xi = 0.01$. For **semantic segmentation**, we evaluate PSL on Cityscapes (Cordts et al., 2016), which contains $2,975$ images for training, and 500 images for testing. The initial labelled pool size is set as $N_L = 30$, the budget is $b = 30$, and the final labelled pool size is $N_f = 180$. We use DRN (Yu et al., 2017) for the task learner and MobileNetV3 (Howard et al., 2019) for the active learner. Mean IoU is used as the evaluation metric. The hyperparameters $\alpha$, $\tau$, and $\xi$ are the same as those in image classification. We train all models for 50 epochs. If not otherwise stated, we use two peers in all our experiments and the reported results are averaged over five individual experimental runs. For fair comparison, we always use the same initial pools. We implement our method with Pytorch (Paszke et al., 2017).

## 4.3 Active Learning in Sensitive Protection Setting

We compare PSL with (I) uncertainty-based methods, Entropy (Shannon, 1948), LL4AL (Yoo & Kweon, 2019) and VAAL (Sinha et al., 2019); and (II) diversity-based methods, Core-set (Sener & Savarese, 2017), BADGE (Ash et al., 2020) and ALFA-Mix (Parvaneh et al., 2022). We also report the performance of a simple random sampling baseline. We present the model complexity in Table 1, where our PSL has **the smallest model size** despite involving multiple peers. In addition, our PSL is also much more computationally efficient than other methods. We compare the sampling time that different methods use to sample the same amount of data from the same unlabelled pool in CIFAR10 (Krizhevsky, 2009) in the sensitive protection setting in Table 2. All experiments are conducted on the same NVIDIA GeForce GTX 1080 Ti GPU for fair comparison. The results demonstrate that our PSL is much more efficient in sampling using the light-weight peer students as the active learner.

**Results on CIFAR10/100** As shown in Figure 3(a)-(b), our PSL outperforms all other methods on CIFAR10/100. On CIFAR10, all methods start with the initial labelled pool that produces the accuracy of $40.8\%$. With 1000 labelled data, PSL is still comparable with VAAL and ALFA-Mix. But when the labelled pool gets larger, PSL starts to outperform the rest methods with a visible margin. PSL achieves $78.06\%$ with the eventual 3000 labelled data. The performance gap between PSL and the second best method LL4AL is around $1.5\%$. And there is a gap of $4.3\%$ between PSL and the lowest results produced by random sampling. On CIFAR100, the initial labelled pool produces the accuracy of $9.1\%$ for all methods. Similar with the situation on CIFAR10, PSL is close to other methods with 1000 labels and starts to show the superiority with more data annotated. PSL finally achieves $28.8\%$ with 3000 labelled data. The performance gap between

Table 2: Comparing the sampling time needed for different methods.

| Model | Time (sec) |
|---|---|
| Entropy (Shannon, 1948) | 0.9 |
| BADGE (Ash et al., 2020) | 17.2 |
| ALFA-Mix (Parvaneh et al., 2022) | 9.1 |
| Core-set (Sener & Savarese, 2017) | 1.2 |
| LL4AL (Yoo & Kweon, 2019) | 8.9 |
| VAAL (Sinha et al., 2019) | 2.5 |
| PSL (ours) | 0.001 |

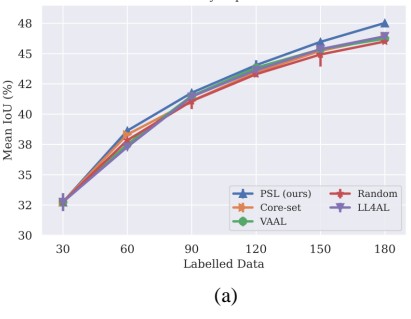

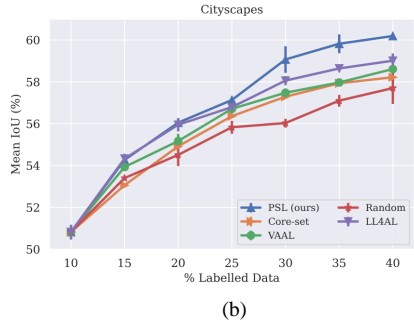

(a)                                        (b)

Figure 4: (a) Active learning in the sensitive protection setting on Cityscapes. The x-axis shows the counts of labelled data. (b) Active learning in the standard setting on Cityscapes. The x-axis shows the percentages of labelled data. The error bars represent the standard deviations among results from multiple different runs.

PSL and the second best method BADGE is around 0.9%. And there is a gap of 4.4% between PSL and the lowest results produced by random sampling.

**Results on Tiny ImageNet**  As shown in Figure 3(c), our PSL outperforms all other methods and achieves 17.3% accuracy with 6,000 labelled data used. A large performance drop is observed for LL4AL, possibly because LL4AL maintains an extra loss learning module which not only trains for sampling but also updates the task learner. The limited data and huge classes might have misled the task learner through the learning module, causing the performance to be even worse than random sampling. We also notice that the random sampling baseline achieves similar results with Core-set and VAAL, possibly because Core-set relies on the quality of extracted features and VAAL relies on the training quality of VAE, which may not generalize well with only tens of annotations for each class. The results demonstrate the ability of PSL on a challenging large-scale dataset.

**Results on Cityscapes**  As shown in Figure 4(a), PSL outperforms all other methods with a clear margin. All methods start with mean IoU of 32.7% using 30 labelled data. The margin between PSL and the rest methods becomes more significant when 150 data are labelled. Eventually with 180 labelled data, PSL achieves 47.5%. Compared to the second best method LL4AL, PSL improves the results by 1.1%.

### 4.4  Active Learning in Standard Setting

Apart from the sensitive protection setting of active learning, we also conduct experiments in the standard active learning setting where no data is protected. We use 10% of the whole dataset as the initial labelled pool and set the budget as 5% of the whole dataset. Again, we conduct the experiments on CIFAR10/100, Tiny Imagenet, and Cityscapes. We reran the models with ResNet-18 backbone for classification and DRN backbone for segmentation under the same settings for a fair comparison. Results are shown in Figure 3(d)-(f) and Figure 4(b). In the standard active learning setting, our PSL still outperforms all the rest methods, demonstrating that our PSL is suitable for not only the sensitive protection setting, but also the standard

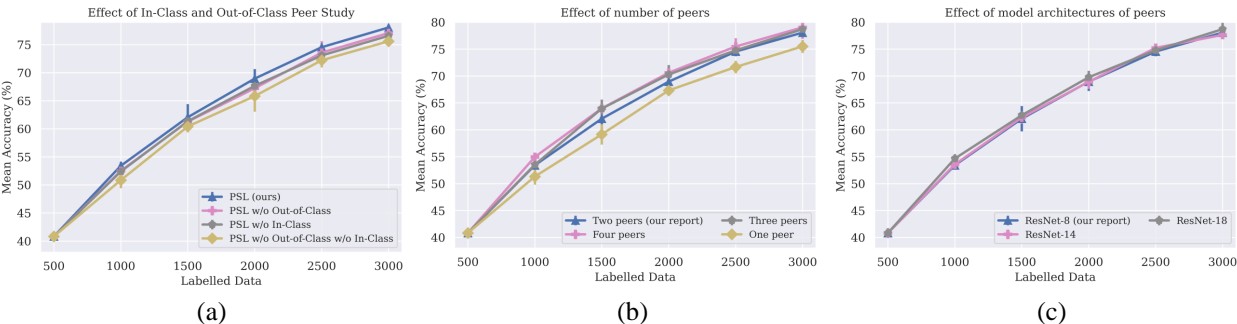

Figure 5: Ablation studies conducted in the sensitive protection setting on CIFAR10.

setting of active learning. We also want to clarify that, although the standard setting for active learning does not protect specific data in the unlabelled pool as the sensitive protection setting does, our PSL still to some extent protects the data privacy by only uploading partial data to the cloud and keeping the rest to the local device.

## 4.5 Ablation Studies

**Effect of In-Class and Out-of-Class Peer Study**    In Figure 5(a), we analyze the effect of In-Class and Out-of-Class Peer Study on CIFAR10. We show the results of PSL, PSL without Out-of-Class Peer Study, PSL without In-Class Peer Study, and PSL without both In-Class and Out-of-Class Peer Study. Specifically, we observe a clear performance drop, around 1.5%, without using Out-of-Class Peer Study or In-Class Peer Study. Without both In-Class and Out-of-Class Peer Study, the performance drop is around 2.5%. These results demonstrate that both In-Class and Out-of-Class Peer Study are crucial to performance improvement in PSL. In-Class Peer Study and Out-of-Class Peer Study depend on and supplement each other in the whole process. In-Class Peer Study provides a better base for the light-weight peers. The randomness added to peers in this process also prepares for the later sampling procedure. Out-of-Class Peer Study, on the other hand, introduces a ranking strategy to improve the sampling results, further making up for the deficiency of light-weight peers. Then Peer Study Feedback actively samples data for annotation. All together make it possible to allow responsible active learning while benefiting from cloud resources.

**Effect of multiple peers**    In Figure 5(b), we study the effect of different number of peer students. Specifically, we show the results using two, three, or four peer students. We also include the result of a single student, i.e., the Peer Study Feedback sampling criterion degrades to the entropy scores. It is observed that two peer students achieve a good trade-off between performance and the costs of computation and memory, while increasing the number of peer students brings further performance improvement. The performance improvement is larger when the labelled data pool is relatively small. Extra peers benefit the model performance with the cost of memory and computation cost. In real-world applications, the number of peers to use can be left to end users to decide.

**Effect of peer model architectures**    Since we use light-weight peer students as active learner, naturally raises the question of using much stronger students. Aside from ResNet-8 (#parameters=0.078M ), we also present the results for using ResNet-14 (#parameters=0.18M) and ResNet-18 (#parameters=11.17M) as student networks in Figure 5(c). We observe that ResNet-14 produces similar results compared to ResNet-8, and ResNet-18 outperforms the other two methods with a small margin. This demonstrates that using ResNet-8 as peer students is a cost-efficient choice. The improvement from using ResNet-18 for peer students is costly, with the model size 143 times of ResNet-8's size and 62 times of ResNet-14's size.

**Effect of a noisy oracle**    In Figure 6, we explore the robustness of our PSL against a noisy oracle, i.e., noisy labels are included. To better model real-world label noise in human annotations, we add 10%, 20%, and 30% label noise to both the labelled pool and the oracle annotations in CIFAR100. Since the 100

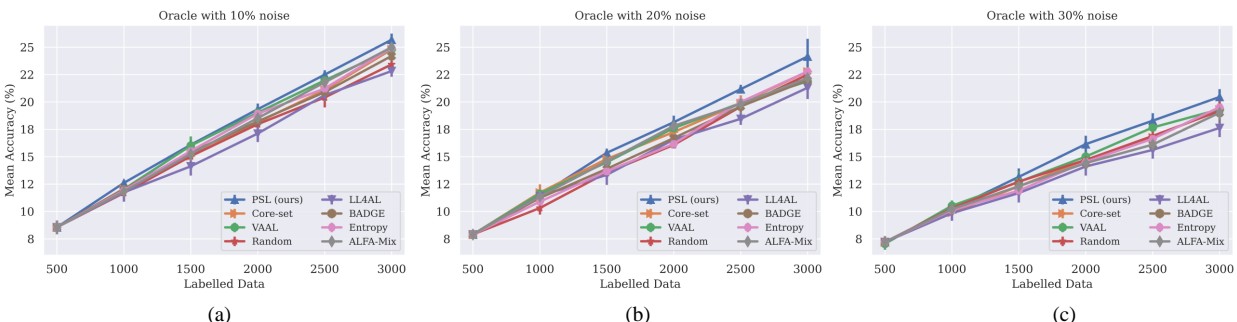

Figure 6: The results from using noisy oracles on CIFAR100. The noisy oracles with 10%, 20%, 30% noise are used respectively.

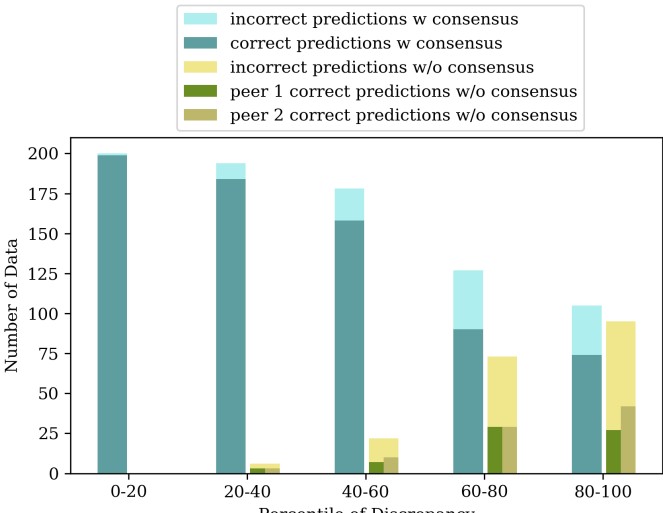

Figure 7: A further justification of Out-of-Class Peer Study on CIFAR10. Specifically, data that reach consensus between peers and data that do not reach consensus between peers are shown in blue and green respectively. Darker colours represent correct predictions and lighter colours represent incorrect predictions.

classes in CIFAR100 can be grouped into 20 superclasses, it is possible to add noise within each superclass to resemble real-world scenarios where similar classes may confuse the annotators. We follow the experiment design of Sinha et al. (2019). We observe that the performances drop for all methods with all three degrees of label noise. And of course, with a larger degree of label noise, the performance becomes worse. Our PSL outperforms all other methods with a clear margin.

**Justification of Out-of-Class Peer Study** To further justify the effectiveness of Out-of-Class Peer Study, we also perform the following experiments. As shown in Figure 7, we train the model with In-Class Peer Study using $4,000$ data and test it with $1,000$ data on CIFAR10. We sort the testing data's discrepancies between peer students 1 and 2 in ascending order and divide them evenly into five groups. In each group, the predictions with consensus always have a higher accuracy than the predictions disagreed between students. This indicates that when we sample with discrepancy, most of the data selected are those cannot be correctly predicted and can improve our target network. Our Out-of-Class Peer Study aims at pushing the data with predictions reaching consensus among students to have a lower discrepancy, and thus making these data less likely to be selected during active sampling. Meanwhile, we increase the model discrepancy for the data with no consensus between students, to increase their active sampling priority.

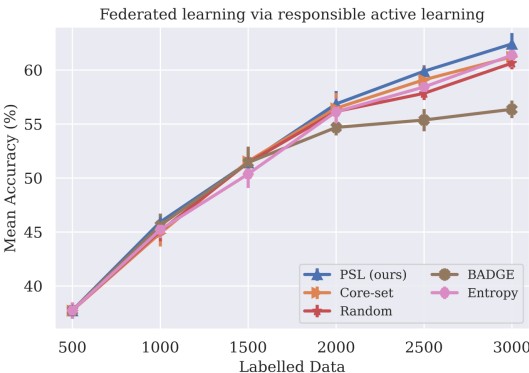

Figure 8: The results of federated learning via active learning on CIFAR10. Multiple local users/clients that locate separately are involved in this setting.

**Federated learning via responsible active learning**   As we mentioned in a previous part of this paper (subsec 3.5), we could fit our responsible active learning in the federated learning framework. Multiple local users are involved in this setting where each of them separately stores part of the training data, e.g., each user can be a photo album on a personal device, and data of different users can be utilized securely and efficiently to build a powerful model on the cloud. Note that this differs from the previous multiple-peer experiment in Subsection 4.5 which involves only one local device despite having multiple peers as the active learner. To run the experiments, we simply split CIFAR10 into random splits to distribute on 10 local clients. On each client, we protect 90% data just like in the sensitive protection setting. Each local client is composed of one active learner (peer students), and one helper model that can be uploaded for aggregation. A task model on the server is maintained to aggregate the local helper models. The peer student, the helper models, and the task model all have the same architecture, ResNet8. The peer students can access all training data and they conduct active sampling. The helper model is trained with only the data sampled from the non-protected set that are considered to be less sensitive. All helper models are uploaded and aggregated using FedAvg (McMahan et al., 2017). The aggregated model is then downloaded to initialize each helper model. We train each client for 40 epochs in each communication round and run for 50 communication rounds in total. This federated learning via responsible active learning framework allows us to strictly protect the extremely sensitive data and also avoid uploading raw training data. We run this experiment with our PSL and several state-of-the-art baselines plus random sampling. We show the results in Figure 8. Our PSL outperforms the other baselines. BADGE performs quite poorly in this setting. Possibly because the limited data on each client makes it difficult to utilize the estimated gradients. When we compare these results to the results from sensitive protection setting shown in Figure 3(a), we find that the results are poorer. This is due to the decentralized training on separated data. More could be explored on the combination of federated learning and responsible active learning, such as the non-IID data distributions across clients. However, these are out of the scope of this work, we will leave it as the subject of future study.

## 5   Conclusion

In this paper, we proposed a new Peer Study Learning framework for responsible active learning, which can simultaneously preserve data privacy and improve model stability. To protect the user data, a teacher-student architecture is introduced to isolate unlabelled data from the task learner by maintaining an active learner for active sampling. In addition, a new learning paradigm is devised to mimic the real-world in-class and out-of-class scenarios. Extensive evaluations have been conducted on both sensitive protection setting and standard active learning benchmarks, where our proposed PSL achieves superior results over a wide range of state-of-the-art active learning methods. We also demonstrated model robustness with a noisy oracle and provided detailed component analysis to validate the insights of our model design.

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
