# OpenReview forum: "Responsible Active Learning via Human-in-the-loop Peer Study"
_TMLR — Rejected by TMLR_

### Review · Reviewer_aNCX · 2022-12-06

**Summary Of Contributions:**

The paper introduces an active learning algorithm in which there exists a single learner (called "teacher") and multiple peers (called "students"), with each student having its own subset of the data. At a high-level, there are three stages:

- *In-Class*: the teacher learns from the labeled data and students distill the knowledge of the teacher as well as the knowledge of their peers.

- *Out-of-class*: The students minimize some chosen loss function that depends on the discrepancy between students.

- *Active Learning*: students select examples using disagreement in a typical active learning setup.

The authors conduct some experiments to argue that the proposed method performs better than other baselines.

**Audience:**

No

**Claims And Evidence:**

No

**Requested Changes:**

* The motivation behind the "out-of-class" learning is unclear. The authors try to explain it by explaining why out of class collaboration is useful, but that has nothing to do with explaining the chosen loss in Equation 5.
* Active learning methods are independent of the choice of the architecture. If the authors would like to add a comparison of size and inference time, please use the same architecture. Of course, that should immediately imply that the proposed algorithm would not have "the best inference time" since the algorithm is far more complicated than a simple uncertainty estimation method using the entropy.
* To claim privacy guarantees, please show that the algorithm satisfies differential privacy. There is no privacy guarantees in the proposed algorithm, contrary to what is claimed in the abstract.
* Please release a code that can reproduce the results.
* There are many typos and complicated mathematical expressions as mentioned above.


**Strengths And Weaknesses:**

*Strengths*: The main strength is that the empirical results show an improvement compared to several baselines. The authors also include ablation studies to demonstrate the impact of each stage in the proposed algorithm.

*Weaknesses*: There are several weaknesses in the paper. First, the authors make some substantial claims, but upon a closer inspection, such claims are unfounded. For example, it is claimed (including in the abstract) that the paper proposes a *responsible* active learning algorithm, which protects privacy. However, this is not really an algorithm that satisfies a privacy guarantee, such as differential privacy. Instead, the authors only say that people can choose which examples are included in the pool in their algorithm to protect their privacy (which is trivial)!

Second, the discussion is confusing in several parts, mainly because the authors insist on drawing analogies with schools, but those analogies can be confusing. For example, the disagreement-based sampling is called "peer study feedback". Also, the "out-of-class" stage is ad-hoc and I don't get the intuition behind that particular formulation. Why is minimizing $|d_D - d_A|$ meaningful? To clarify, $d_D$ is the average level of disagreement among students on examples in which they already disagree, while $d_A$ is the average level of disagreement on examples in which they agree. What would be plausible is to minimize the overall disagreement, but then that could hurt during the sampling step. It also doesn't help that the authors write something as simple as an absolute difference $|x-y|$ using the expression $\mathbb{I}(x, y)(x-y)$, where $\mathbb{I}(x, y) = 1$ if $x>y$ and $\mathbb{I}(x, y) = -1$ otherwise. The whole complicated expression in Eq (5) is just $|x-y|$.

Third, the paper would benefit from proofreading. There are many typos including in prominent places, like the algorithm and the main experiment's figure. For example, the paper writes "Equation X" using "Eq. equation X". Also, in Figure 3, the authors refer in the caption to a forth (removed?) column that is not in the figure.

Finally, the paper makes confusing remarks in the experiments section. For example, the authors say that their algorithm "has" the smallest architecture size, but the architecture is independent of the choice of the active learning algorithm.  If you use MobileNetV3 in your method, you can use MobielNetV3 when running the other active learning algorithms, such as using uncertainty sampling. The authors use MobileNetV3 and claim after that their method has the least model size and the smallest inference time!

---

### Review · Reviewer_ULLe · 2022-12-11

**Summary Of Contributions:**

The paper identified the potential privacy problem when uploading unlabeled data to the cloud side for active learning. They introduced a new sensitive protection setting for trustworthy active learning and proposed the corresponding solution framework Peer Study Learning (PSL). Inspired by real-world student learning, they designed three modules: (1) In-Class Peer Study, (2) Out-of-Class Peer Study, and (3) Peer Study Feedback to achieve responsible active learning. For In-Class Peer Study, the peer students will learn under teacher guidance. Then the students collaboratively learn in a group on unlabeled data without teacher guidance in Out-of-Class Peer Study. Finally, the discrepancy of peer students is used as feedback to select the most informative samples for active learning. The framework is argued to do data isolation and thus protect privacy. The empirical experiments are conducted in both standard and sensitive protection settings.

**Audience:**

Yes

**Broader Impact Concerns:**

I think the paper may need to illustrate more about sensitive information protection in active learning.

**Claims And Evidence:**

Yes

**Requested Changes:**

I hope the authors could answer all the questions mentioned in the above weaknesses. Among them, I think the privacy problem and the missing experimental setups are my major concerns. Addressing them will strengthen the work.

**Strengths And Weaknesses:**

For strengths,
-	The paper is well-written and flows smoothly.
-	The introduced problem is meaningful and has real-world implications. The motivation for studying responsible active learning is also clear.
-	The peer study learning framework is interesting, especially leveraging the peer feedback discrepancy as the metric to select informative samples to be annotated. The out-of-class peer study and peer study feedback modules are also well integrated into the framework for better sample selection. I think these can bring insights to the community.
-	The experiments are conducted in various setups. The results of both the standard and sensitive protection settings are appreciated.
For weaknesses,
-	Privacy is a major issue. As the paper mentioned, the model will leverage the protected data during the out-of-class study stage. I think there is a risk of privacy information leakage across different participating users here. Moreover, the sensitive protection in the paper denotes selecting samples from the remaining 10% unprotected images. However, in the actual experiment, 10% of the unprotected data was just randomly selected, which was oversimplified. The problem of how to define sensitive protection was not well solved, and it may also be unrealistic to rely only on users to select. This part requires further justifications.
-	The paper said there are multiple peer students for collaborative learning. How many peer students are used in the actual experiments and how are the students defined in the image classification data sets? I found in the out-of-class peer study stage, the data selection and optimization will require the involvement of all the student models. Therefore, I’m curious about the scalability of the proposed framework for a large number of users/students. More experimental details should be put into the paper.
-	From my understanding, the framework needs to communicate the data, labels, and predictions between the cloud and the students. Could you explain the communication cost here? I’m not sure if it’s acceptable.
-	For the experiments, despite the comparisons of the sampling time, I’d also like to know the complete training time comparisons with the baselines.
-	Why the baseline selection is different for different experimental setups? For example, the baselines in Figures 3 (c), (f) are different from Figures 3 (a), (b), (d), (e). Then the baselines in Figures 4, 6, and figure 8. The reasons need to be explained.

---

### Review · Reviewer_oRwo · 2023-02-09

**Summary Of Contributions:**

This paper argued that traditional active learning setting faces the challenge of the data privacy leakage issue. To preserve data privacy and improve model stability, this paper further proposed a responsible active learning method, where a set of offline student models select the data to be uploaded for annotation based on their discrepancy on the unlabeled samples. This work further simulated a sensitive protection setting and conducted experiments on CIFAR and Tiny ImageNet.

**Audience:**

Yes

**Broader Impact Concerns:**

I currently have no concerns about the ethical implications of the work.

**Claims And Evidence:**

Yes

**Requested Changes:**

Please see the weaknesses for the requested changes.

In Figure 2, the in-class peer study does not include how the teacher model is trained (i.e., using ground-truth annotations) and how the student models learn from ground-truth labels.

**Strengths And Weaknesses:**

Strengths

* The newly proposed responsible active learning combines data privacy preservation and active learning, which is more practical.

* Based on the simulation, the proposed method achieved comparable performance with a significantly small model and less sampling time. The experiments are conducted on both classification and semantic segmentation.

Weaknesses

* The sensitive protection simulation is questionable. As stated in Section 4.1, "we randomly select 90% data as the protected set, then the training starts with a labeled pool containing NL images from the non-protected set". However, the selection of uploaded data and the informativeness of data are supposed to be correlated. In other words, the uploaded data may have many common characteristics so that it did not contain much privacy, but the informative data are more likely to include sensitive personal information. Therefore, the confounder of data selection and data information is ignored in the simulation. If so, this issue would challenge the correction of the simulation.

* The main contribution of this work is the new concept of "responsible active learning". However, I didn't find a clear definition of "responsible", which confused me in understanding the background and contributions of this work.

* Page 2 Line 12 claimed that previous active sampling solutions "rely on representations learned by a single model, potentially making the sampling prone to bias". However, I didn't see further discussions or evidence on why the bias was introduced. As this motivates the design of multiple peer students, it is important to include a detailed analysis of the bias from a single model.

* It is strange to me why the task learner is necessary to the framework. The task learner is used to generate soft labels as additional supervision for the offline model, and is independent of the sample selection process. Therefore, the task learner is just like using knowledge distillation to get better performance. If so, the task learner is more like a trick but redundant. I would like to know what is the role of the task learner in responsible active learning.

* As there are multiple models working as the student, it is not clear whether all the models are used during inference. I am wondering whether the performance gain comes from the ensemble of models, and whether compared baselines can achieve comparable performance with a similar ensemble architecture.

---

### Decision · Action_Editors · 2023-03-29

**Recommendation:** Reject

**Comment:**

All the reviewers acknowledged the novelty of the paper---it is indeed a pioneering paper that studies the privacy issue in active learning. However, as I summarized in the first section, the three key concerns are not successfully addressed after the rebuttal.
All the reviewers vote for rejection.

**Audience:**

Yes

**Claims And Evidence:**

The papers have the following key claims that raise considerable concerns from all the reviewers:
1) Protected and unprotected data are IID. However, this is not realistic. For example, intimate friend photos (private) certainly differ from food photos (public).
2) This method is essentially formulated in a distillation teacher-student view, how does it relate to responsibility? The authors failed to justify it.
3) The human factor analysis is problematic.